# Understanding Why Neural Networks Generalize Well Through GSNR of Parameters

Jinlong Liu[1,*], Guo-qing Jiang[1], Yunzhi Bai[1], Ting Chen[2], and Huayan Wang[1]

[1]Ytech – KWAI incorporation
{liujinlong,jiangguoqing,baiyunzhi,wanghuayan}@kuaishou.com
[2]Samsung Research China – Beijing (SRC-B)
ting11.chen@samsung.com

## Abstract

As deep neural networks (DNNs) achieve tremendous success across many application domains, researchers tried to explore in many aspects on why they generalize well. In this paper, we provide a novel perspective on these issues using the gradient signal to noise ratio (GSNR) of parameters during training process of DNNs. The GSNR of a parameter is defined as the ratio between its gradient's squared mean and variance, over the data distribution. Based on several approximations, we establish a quantitative relationship between model parameters' GSNR and the generalization gap. This relationship indicates that larger GSNR during training process leads to better generalization performance. Moreover, we show that, different from that of shallow models (e.g. logistic regression, support vector machines), the gradient descent optimization dynamics of DNNs naturally produces large GSNR during training, which is probably the key to DNNs' remarkable generalization ability.

## 1 Introduction

Deep neural networks typically contain far more trainable parameters than training samples, which seems to easily cause a poor generalization performance. However, in fact they usually exhibit remarkably small generalization gaps. Traditional generalization theories such as VC dimension (Vapnik & Chervonenkis, 1991) or Rademacher complexity (Bartlett & Mendelson, 2002) cannot explain its mechanism. Extensive research focuses on the generalization ability of DNNs (Neyshabur et al., 2017; Arora et al., 2018; Keskar et al., 2016; Dinh et al., 2017; Hoffer et al., 2017; Novak et al., 2018; Dziugaite & Roy, 2017; Jakubovitz et al., 2019; Kawaguchi et al., 2017; Advani & Saxe, 2017).

Unlike that of shallow models such as logistic regression or support vector machines, the global minimum of high-dimensional and non-convex DNNs cannot be found analytically, but can only be approximated by gradient descent and its variants (Zeiler, 2012; Kingma & Ba, 2014; Graves, 2013). Previous work (Zhang et al., 2016; Hardt et al., 2015; Dziugaite & Roy, 2017) suggests that the generalization ability of DNNs is closely related to gradient descent optimization. For example, Hardt et al. (2015) claims that any model trained with stochastic gradient descent (SGD) for reasonable epochs would exhibit small generalization error. Their analysis is based on the smoothness of loss function. In this work, we attempt to understand the generalization behavior of DNNs through GSNR and reveal how GSNR affects the training dynamics of gradient descent. Stanislav Fort (2019) studied a new gradient alignment measure called stiffness in order to understand generalization better and stiffness is related to our work.

The GSNR of a parameter is defined as the ratio between its gradient's squared mean and variance over the data distribution. Previous work tried to use GSNR to conduct theoretical analysis on deep learning. For example, Rainforth et al. (2018) used GSNR to analyze variational bounds in

---

*corresponding author

unsupervised DNNs such as variational auto-encoder (VAE). Here we focus on analyzing the relation between GSNR and the generalization gap.

Intuitively, GSNR measures the similarity of a parameter's gradients among different training samples. Large GSNR implies that most training samples agree on the optimization direction of this parameter, thus the parameter is more likely to be associated with a meaningful "pattern" and we assume its update could lead to a better generalization. In this work, we prove that the GSNR is strongly related to the generalization performance, and larger GSNR means a better generalization.

To reveal the mechanism of DNNs' good generalization ability, we show that the gradient descent optimization dynamics of DNN naturally leads to large GSNR of model parameters and therefore good generalization. Furthermore, we give a complete analysis and a detailed interpretation to this phenomenon. We believe this is probably the key to DNNs remarkable generalization ability.

In the remainder of this paper we first analyze the relation between GSNR and generalization (Section 2). We then show how the training dynamics lead to large GSNR of model parameters experimentally and analytically in Section 3.

## 2 LARGER GSNR LEADS TO BETTER GENERALIZATION

In this section, we establish a quantitative relation between the GSNR of model parameters and generalization gap, showing that larger GSNR during training leads to better generalization.

### 2.1 GRADIENTS SIGNAL TO NOISE RATIO

Consider a data distribution $\mathcal{Z} = \mathcal{X} \times \mathcal{Y}$, from which each sample $(x, y)$ is drawn; a model $\hat{y} = f(x, \theta)$ parameterized by $\theta$; and a loss function $L$.

The parameters' gradient *w.r.t.* $L$ and sample $(x_i, y_i)$ is denoted by

$$\mathbf{g}(x_i, y_i, \theta) \ \text{or} \ \mathbf{g}_i(\theta) := \frac{\partial L(y_i, f(x_i, \theta))}{\partial \theta} \tag{1}$$

whose $j$-th element is $\mathbf{g}_i(\theta_j)$. Note that throughout this paper we always use $i$ to index data examples and $j$ to index model parameters.

Given the data distribution $\mathcal{Z}$, we have the (sample-wise) mean and variance of $\mathbf{g}_i(\theta)$. We denote them as $\tilde{\mathbf{g}}(\theta) = \mathrm{E}_{(x,y)\sim\mathcal{Z}}(\mathbf{g}(x, y, \theta))$ and $\rho^2(\theta) = \mathrm{Var}_{(x,y)\sim\mathcal{Z}}(\mathbf{g}(x, y, \theta))$, respectively.

The gradient signal to noise ratio (GSNR) of one model parameter $\theta_j$ is defined as:

$$r(\theta_j) := \frac{\tilde{\mathbf{g}}^2(\theta_j)}{\rho^2(\theta_j)} \tag{2}$$

At a particular point of the parameter space, GSNR measures the consistency of a parameter's gradients across different data samples. Figure 1 intuitively shows that if GSNR is large, the parameter gradient space tends to be distributed in the similar direction and if GSNR is small, the gradient vectors are then scatteredly distributed.

### 2.2 ONE-STEP GENERALIZATION RATIO

In this section we introduce a new concept to help measure the generalization performance during gradient descent optimization, which we call one-step generalization ratio (OSGR). Consider training set $D = \{(x_1, y_1), ..., (x_n, y_n)\} \sim \mathcal{Z}^n$ with $n$ samples drawn from $\mathcal{Z}$, and a test set $D' = \{(x'_1, y'_1), ..., (x'_{n'}, y'_{n'})\} \sim \mathcal{Z}^{n'}$. In practice we use the loss on $D'$ to measure generalization. For simplicity, we assume the sizes of training and test datasets are equal, *i.e.* $n = n'$. We denote the empirical training and test loss as:

$$L[D] = \frac{1}{n}\sum_{i=1}^{n} L(y_i, f(x_i, \theta)), \quad L[D'] = \frac{1}{n}\sum_{i=1}^{n} L(y'_i, f(x'_i, \theta)), \tag{3}$$

respectively. Then the empirical generalization gap is given by $L[D'] - L[D]$.

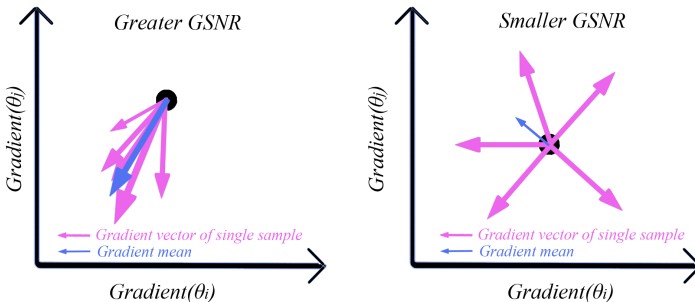

Figure 1: Schematic diagram of the sample-wise parameter gradient distribution corresponding to greater (**Left**) and smaller (**Right**) GSNR. Pink arrows denote the gradient vectors for each sample while the blue arrow indicates their mean.

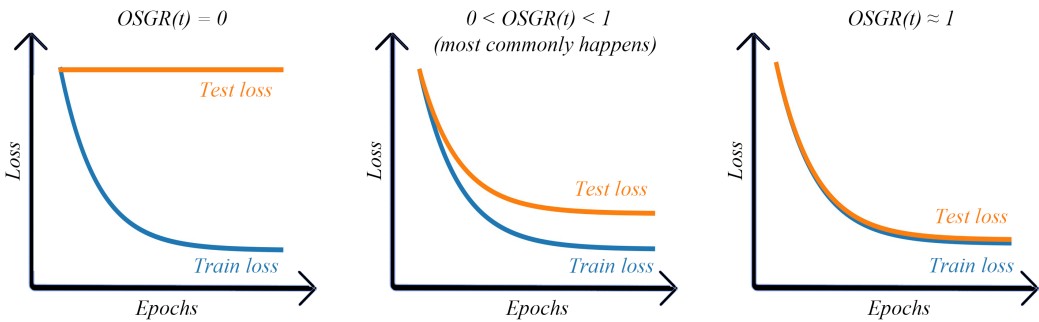

Figure 2: Schematic diagram of the training behavior satisfies $OSGR(t) = 0$ (**Left**), $0 < OSGR(t) < 1$ (**Middle**) and $OSGR(t) \approx 1$ (**Right**). Note that the **Middle** scenario most commonly happens in regular tasks.

In gradient descent optimization, both the training and test loss would decrease step by step. We use $\Delta L[D]$ and $\Delta L[D']$ to denote the one-step training and test loss decrease during training, respectively. Let's consider the ratio between the expectations of $\Delta L[D']$ and $\Delta L[D]$ of one single training step, which we denote as $\mathbf{R}(\mathcal{Z}, n)$.

$$\mathbf{R}(\mathcal{Z}, n) := \frac{E_{D,D' \sim \mathcal{Z}^n}(\Delta L[D'])}{E_{D \sim \mathcal{Z}^n}(\Delta L[D])} \tag{4}$$

Note that this ratio also depends on current model parameters $\theta$ and learning rate $\lambda$. We are not including them in the above notation as we will not explicitly model these dependencies, but rather try to quantitatively characterize $\mathbf{R}$ for very small $\lambda$ and for $\theta$ at the early stage of training (satisfying Assumption 2.3.1).

Also note that the expectation of $\Delta L[D']$ is over $D$ and $D'$. This is because the optimization step is performed on $D$. We refer to $\mathbf{R}(\mathcal{Z}, n)$ as OSGR of gradient descent optimization. Statistically the training loss decreases faster than the test loss and $0 < OSGR(t) < 1$ (**Middle** panel of Figure 2), which usually results in a non-zero generalization gap at the end of training. If $OSGR(t)$ is large ($\approx 1$) in the whole training process (**Right** panel of Figure 2), generalization gap would be small when training completes, implying good generalization ability of the model. If $OSGR(t)$ is small ($= 0$), the test loss will not decrease while the training loss normally drops (**Left** panel of Figure 2), corresponding to a large generalization gap.

## 2.3 RELATION BETWEEN GSNR AND OSGR

In this section, we derive a relation between the OSGR during training and the GSNR of model parameters. This relation indicates that, for the first time as far as we know, the sample-wise gradient distribution of parameters is related to the generalization performance of gradient descent optimization.

In gradient descent optimization, we take the average gradient over training set $D$, which we denote as $\mathbf{g}_D(\theta)$. Note that we have used $\mathbf{g}_i(\theta)$ to denote gradient evaluated on one data sample and $\tilde{\mathbf{g}}(\theta)$ to denote its expectation over the entire data distribution. Similarly we define $\mathbf{g}_{D'}(\theta)$ to be the average gradient over test set $D'$.

$$\mathbf{g}_D(\theta) = \frac{1}{n}\sum_{i=1}^{n}\mathbf{g}(x_i, y_i, \theta) = \frac{\partial L[D]}{\partial \theta} \quad , \quad \mathbf{g}_{D'}(\theta) = \frac{1}{n}\sum_{i=1}^{n}\mathbf{g}(x'_i, y'_i, \theta) = \frac{\partial L[D']}{\partial \theta} \tag{5}$$

Both the training and test dataset are randomly generated from the same distribution $\mathcal{Z}^n$, so we can treat $\mathbf{g}_D(\theta)$ and $\mathbf{g}_{D'}(\theta)$ as random variables. At the beginning of the optimization process, $\theta$ is randomly initialized thus independent of $D$, so $\mathbf{g}_D(\theta)$ and $\mathbf{g}_{D'}(\theta)$ would obey the same distribution. After a period of training, the model parameters begin to fit the training dataset and become a function of $D$, i.e. $\theta = \theta(D)$, therefore distributions of $\mathbf{g}_D(\theta(D))$ and $\mathbf{g}_{D'}(\theta(D))$ become different. However we choose not to model this dependency and make the following assumption for our analysis:

**Assumption 2.3.1 (Non-overfitting limit approximation)** *The average gradient over the training dataset and test dataset $\mathbf{g}_D(\theta)$ and $\mathbf{g}_{D'}(\theta)$ obey the same distribution.*

Obviously the mean of $\mathbf{g}_D(\theta)$ and $\mathbf{g}_{D'}(\theta)$ is just the mean gradient over the data distribution $\tilde{\mathbf{g}}(\theta)$.

$$\mathrm{E}_{D\sim\mathcal{Z}^n}[\mathbf{g}_D(\theta)] = \mathrm{E}_{D,D'\sim\mathcal{Z}^n}[\mathbf{g}_{D'}(\theta)] = \tilde{\mathbf{g}}(\theta) \tag{6}$$

We denote their variance as $\sigma^2(\theta)$, i.e.

$$\mathrm{Var}_{D\sim\mathcal{Z}^n}[\mathbf{g}_D(\theta)] = \mathrm{Var}_{D,D'\sim\mathcal{Z}^n}[\mathbf{g}_{D'}(\theta)] = \sigma^2(\theta) \tag{7}$$

It is straightforward to show that:

$$\sigma^2(\theta) = \mathrm{Var}_{D\sim\mathcal{Z}^n}[\frac{1}{n}\sum_{i=1}^{n}\mathbf{g}_i(\theta)] = \frac{1}{n}\rho^2(\theta) \tag{8}$$

where $\sigma^2(\theta)$ is the variance of the average gradient over the dataset of size $n$, and $\rho^2(\theta)$ is the variance of the gradient of a single data sample.

In one gradient descent step, the model parameter is updated by $\Delta\theta = \theta_{t+1} - \theta_t = -\lambda\mathbf{g}_D(\theta)$ where $\lambda$ is the learning rate. If $\lambda$ is small enough, the one-step training and test loss decrease can be approximated by

$$\Delta L[D] \approx -\Delta\theta \cdot \frac{\partial L[D]}{\partial\theta} + O(\lambda^2) = \lambda\mathbf{g}_D(\theta)\cdot\mathbf{g}_D(\theta) + O(\lambda^2) \tag{9}$$

$$\Delta L[D'] \approx -\Delta\theta \cdot \frac{\partial L[D']}{\partial\theta} + O(\lambda^2) = \lambda\mathbf{g}_D(\theta)\cdot\mathbf{g}_{D'}(\theta) + O(\lambda^2) \tag{10}$$

Usually there are some differences between the directions of $\mathbf{g}_D(\theta)$ and $\mathbf{g}_{D'}(\theta)$, so statistically $\Delta L[D]$ tends to be larger than $\Delta L[D']$ and the generalization gap would increase during training. When $\lambda \to 0$, in one single training step the empirical generalization gap increases by $\Delta L[D] - \Delta L[D']$, for simplicity we denote this quantity as $\bigtriangledown$:

$$\begin{aligned}
\bigtriangledown := \Delta L[D] - \Delta L[D'] &\approx \lambda\mathbf{g}_D(\theta)\cdot\mathbf{g}_D(\theta) - \lambda\mathbf{g}_D(\theta)\cdot\mathbf{g}_{D'}(\theta) & (11)\\
&= \lambda(\tilde{\mathbf{g}}(\theta)+\epsilon)(\tilde{\mathbf{g}}(\theta)+\epsilon-\tilde{\mathbf{g}}(\theta)-\epsilon') & (12)\\
&= \lambda(\tilde{\mathbf{g}}(\theta)+\epsilon)(\epsilon-\epsilon') & (13)
\end{aligned}$$

Here we replaced the random variables by $\mathbf{g}_D(\theta) = \tilde{\mathbf{g}}(\theta) + \epsilon$ and $\mathbf{g}_{D'}(\theta) = \tilde{\mathbf{g}}(\theta) + \epsilon'$, where $\epsilon$ and $\epsilon'$ are random variables with zero mean and variance $\sigma^2(\theta)$. Since $E(\epsilon') = E(\epsilon) = 0$, $\epsilon$ and $\epsilon'$ are independent, the expectation of $\bigtriangledown$ is

$$E_{D,D'\sim\mathcal{Z}^n}(\bigtriangledown) = E(\lambda\epsilon\cdot\epsilon) + O(\lambda^2) = \lambda\sum_j\sigma^2(\theta_j) + O(\lambda^2) \tag{14}$$

where $\sigma^2(\theta_j)$ is the variance the of average gradient of the parameter $\theta_j$.

For simplicity, when it involves a single model parameter $\theta_j$, we will use only a subscript $j$ instead of the full notation. For example, we use $\sigma_j^2$, $r_j$, and $\mathbf{g}_{D,j}$ to denote $\sigma^2(\theta_j)$, $r(\theta_j)$, and $\mathbf{g}_D(\theta_j)$ respectively.

Consider the expectation of $\Delta L[D]$ and $\Delta L[D']$ when $\lambda \to 0$

$$E_{D\sim\mathcal{Z}^n}(\Delta L[D]) \approx \lambda E_{D\sim\mathcal{Z}^n}(\mathbf{g}_D(\theta) \cdot \mathbf{g}_D(\theta)) = \lambda \sum_j E_{D\sim\mathcal{Z}^n}(\mathbf{g}_{D,j}^2) \tag{15}$$

$$\begin{aligned} E_{D,D'\sim\mathcal{Z}^n}(\Delta L[D']) &= E_{D,D'\sim\mathcal{Z}^n}(\Delta L[D] - \bigtriangledown) \tag{16} \\ &\approx \lambda \sum_j (E_{D\sim\mathcal{Z}^n}(\mathbf{g}_{D,j}^2) - \sigma_j^2) \tag{17} \\ &= \lambda \sum_j (E_{D\sim\mathcal{Z}^n}(\mathbf{g}_{D,j}^2) - \rho_j^2/n) \tag{18} \end{aligned}$$

Substituting (18) and (15) into (4) we have:

$$\mathbf{R}(\mathcal{Z}, n) = 1 - \frac{\sum_j \rho_j^2}{n \sum_j E_{D\sim\mathcal{Z}^n}(\mathbf{g}_{D,j}^2)} \tag{19}$$

Although we derived eq. (19) from simplified assumptions, we can empirically verify it by estimating two sides of the equation on real data. We will elaborate on this estimation method in section 2.4.

We can rewrite eq. (19) as:

$$\begin{aligned} \mathbf{R}(\mathcal{Z}, n) &= 1 - \frac{1}{n} \sum_j \frac{E_{D\sim\mathcal{Z}^n}(\mathbf{g}_{D,j}^2)}{\sum_{j'} E_{D\sim\mathcal{Z}^n}(\mathbf{g}_{D,j'}^2)} \frac{\rho_j^2}{E_{D\sim\mathcal{Z}^n}(\mathbf{g}_{D,j}^2)} \tag{20} \\ &= 1 - \frac{1}{n} \sum_j \frac{E_{D\sim\mathcal{Z}^n}(\mathbf{g}_{D,j}^2)}{\sum_{j'} E_{D\sim\mathcal{Z}^n}(\mathbf{g}_{D,j'}^2)} \frac{1}{r_j + \frac{1}{n}} \tag{21} \end{aligned}$$

where $E_{D\sim\mathcal{Z}^n}(\mathbf{g}_{D,j}^2) = Var_{D\sim\mathcal{Z}^n}(\mathbf{g}_{D,j}) + E_{D\sim\mathcal{Z}^n}^2(\mathbf{g}_{D,j}) = \frac{1}{n}\rho_j^2 + \tilde{\mathbf{g}}_j^2$.

We define $\Delta L_j[D]$ to be the training loss decrease caused by updating $\theta_j$. We can show that when $\lambda$ is very small $\Delta L_j[D] = \lambda \mathbf{g}_{D,j}^2 + O(\lambda^2)$. Therefore when $\lambda \to 0$, we have

$$\mathbf{R}(\mathcal{Z}, n) = 1 - \frac{1}{n} \sum_j W_j \frac{1}{r_j + \frac{1}{n}}, \quad \text{where } W_j := \frac{E_{D\sim\mathcal{Z}^n}(\Delta L_j[D])}{E_{D\sim\mathcal{Z}^n}(\Delta L[D])} \quad \text{with} \sum_j W_j = 1 \tag{22}$$

Eq. (22) shows that the GSNR $r_j$ plays a crucial role in the model's generalization ability—the one-step generalization ratio in gradient descent equals one minus the weighted average of $\frac{1}{r_j + \frac{1}{n}}$ over all model parameters divided by $n$. The weight is proportional to the expectation of the training loss decrease resulted from updating that parameter. This implies that larger GSNR of model parameters during training leads to smaller generalization gap growth thus better generalization performance of the trained model. Also note when $n \to \infty$, we have $\mathbf{R}(\mathcal{Z}, n) \to 1$, meaning that training on more data helps generalization.

## 2.4 EXPERIMENTAL VERIFICATION OF THE RELATION BETWEEN GSNR AND OSGR

The relation between GSNR and OSGR, *i.e.* eq. (19) or (22) can be empirically verified using any dataset if: (1) The dataset includes enough samples to construct many training sets and a large enough test set so that we can reliably estimate $\rho_j^2$, $E_{D\sim\mathcal{Z}^n}(\mathbf{g}_{D,j}^2)$ and OSGR. (2) The learning rate is small enough. (3) In the early training stage of gradient descent.

To empirically verify eq. (19), we show how to estimate its left and right hand sides, *i.e.* OSGR by definition and OSGR as a function of GSNR. Suppose we have $M$ training sets each with size $n$, and a test set of size $n'$. We initialize a model and train it separately on the $M$ training sets and test it with the same test set. For the $t$-th training iteration, we denote the training loss and test loss of the model trained on the $m$-th training dataset as $L_t^{(m)}$ and $L'^{(m)}_t$, respectively. Then the left hand

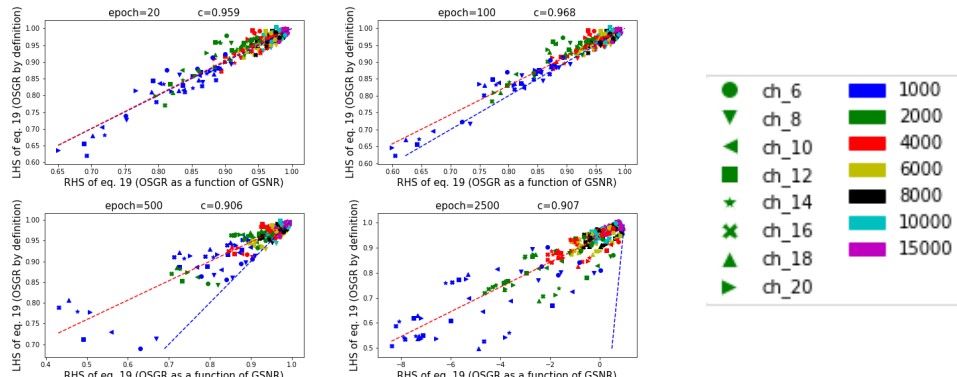

Figure 3: Left hand (LHS or OSGR by definition) and right side (RHS or OSGR as a function of GSNR) of eq. (19). Points are drawn under different experiment settings. **Left**: LHS vs RHS at epoch 20, 100, 500, 2500. Each point is drawn by LHS and RHS computed at the given epoch under different model structure (number of channels) or training data size; red dotted line is the line of best fit computed by least squares; blue dotted line is the line of reference representing LHS = RHS; the value of $c$ in each title represents the Pearson correlation coefficient between LHS and RHS computed by points in figure. **Right**: The legend. Different symbols and colors stand for different number of channels and training data size. Different random noise levels are not distinguished.

side, *i.e.* OSGR by definition, of the $t$-th iteration can be estimated by

$$\mathbf{R}_t(\mathcal{Z}, n) \approx \frac{\sum_{m=1}^{M} L'^{(m)}_{t+1} - L'^{(m)}_t}{\sum_{m=1}^{M} L^{(m)}_{t+1} - L^{(m)}_t} \tag{23}$$

For the model trained on the $m$-th training set, we can compute the $t$-th step average gradient and sample-wise gradient variance of $\theta_j$ on the corresponding training set, denoted as $\mathbf{g}_{m,j,t}$ and $\rho^2_{m,j,t}$, respectively. Therefore the right hand side of eq. (19) can be estimated by

$$E_{D \sim \mathcal{Z}^n}(\mathbf{g}^2_{D,j,t}) \approx \frac{1}{M} \sum_{m=1}^{M} \mathbf{g}^2_{m,j,t}, \quad \rho^2_{j,t} \approx \frac{1}{M} \sum_{m=1}^{M} \rho^2_{m,j,t} \tag{24}$$

We performed the above estimations on MNIST with a simple CNN structure consists of 2 Conv-Relu-MaxPooling blocks and 2 fully-connected layers. First, to estimate eq. (24) with $M = 10$, we randomly sample 10 training sets with size $n$ and a test set with size 10,000. To cover different conditions, we (1) choose $n \in \{1000, 2000, 4000, 6000, 8000, 10000, 15000\}$, respectively; (2) inject noise by randomly changing the labels with probability $p_{random} \in \{0.0, 0.1, 0.2, 0.3, 0.5\}$; (3) change the model structure by varying number of channels in the layers, $ch \in \{6, 8, 10, 12, 14, 16, 18, 20\}$. See Appendix A for more details of the setup. We use the gradient descent training (not SGD), with a small learning rate of $0.001$. The left and right hand sides of 19 at different epochs are shown in Figure 3, where each point represents one specific choice of the above settings.

At the beginning of training, the data points are closely distributed along the dashed line corresponding to LHS=RHS. This shows that eq. (19) fits quite well under a variety of different settings. As training proceeds, the points become more scattered as the non-overfitting limit approximation no longer holds, but correlation between the LHS and RHS remains high even when the training converges (at epoch 2,500). We also conducted the same experiment on CIFAR10 A.2 and a toy dataset A.3 observed the same behavior. See Appendix for these experiments.

The empirical evidence together with our previous derivation of eq. (19) clearly show the relation between GSNR and OSGR and its implication in the model's generalization ability.

# 3 TRAINING DYNAMICS OF DNNS NATURALLY LEADS TO LARGE GSNR

In this section, we analyze and explain one interesting phenomenon: the parameters' GSNR of DNNs rises in the early stages of training, whereas the GSNR of shallow models such as logistic regression or support vector machines declines during the entire training process. This difference gives rise to GSNR's large practical values during training, which in turn is associated with good

generalization. We analyze the dynamics behind this phenomenon both experimentally and theoretically.

## 3.1 GSNR BEHAVIOR OF DNNs TRAINING

For shallow models, the GSNR of parameters decreases in the whole training process because gradients become small as learning converges. But for DNNs it is not the case. We trained DNNs on the CIFAR datasets and computed the GSNR averaged over all model parameters. Because $E_{D \sim \mathcal{Z}^n}(\mathbf{g}_{D,j}^2) = \frac{1}{n}\rho_j^2 + \tilde{\mathbf{g}}_j^2$ and we assume $n$ is large, $E_{D \sim \mathcal{Z}^n}(\mathbf{g}_{D,j}^2) \approx \tilde{\mathbf{g}}_j^2$. In the case of only one large training datasets, we estimate GSNR of $t$-th iteration by

$$r_{j,t} \approx \mathbf{g}_{D,j,t}^2 / \rho_{D,j,t}^2 \tag{25}$$

As shown in Figure 4, the GSNR starts out low with randomly initialized parameters. As learning progresses, the GSNR increases in the early training stage and stays at a high level in the whole learning process. For each model parameter, we also computed the proportion of the samples with the same gradient sign, denoted as $p_{same\_sign}$. In Figure 4c, we plot the mean of time series of this proportion for all the parameters. This value increases from about 50% (half positive half negetive due to random initialization) to about 56% finally, which indicates that for most parameters, the gradient signs on different samples become more consistent. This is because meaningful features begin to emerge in the learning process and the gradients of the weights on these features tend to have the same sign among different samples.

Previous research (Zhang et al., 2016) showed that DNNs achieved zero training loss by memorizing training samples even if the labels were randomized. We also plot the average GSNR for model trained using data with randomized labels in Figure 4 and find that the GSNR stays at a low level throughout the training process. Although the training loss of both the original and randomized labels go to zero (not shown), the GSNR curves clearly distinguish between these two cases and reveal the lack of meaningful patterns in the latter one. We believe this is the reason why DNNs trained on real and random data lead to completely different generalization behaviors.

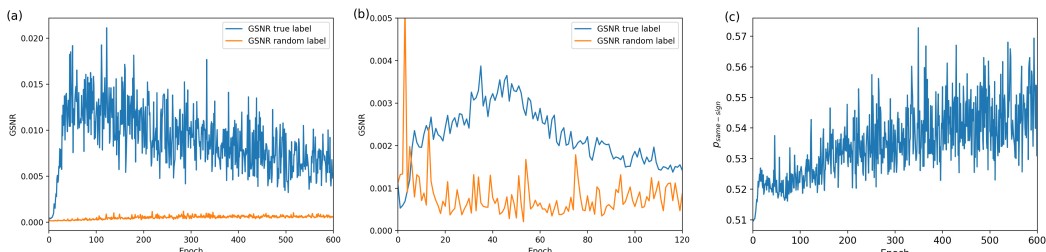

Figure 4: **(a)**: GSNR curves generated by a simple network based on real and random data. An obvious upward process in the early training stage was observed for real data only. **(b)**: Same plot for ResNet18. **(c)**: Average of $p_{same\_sign}$ for the same model as in (a).

## 3.2 TRAINING DYNAMICS BEHIND THE GSNR BEHAVIOR

In this section we show that the feature learning ability of DNNs is the key reason why the GSNR curve behavior of DNNs is different from that of shallow models during the gradient descent training. To demonstrate this, a simple two-layer perceptron regression model is constructed. A synthetic dataset is generated as following. Each data point is constructed *i.i.d.* using $y = x_0 x_1 + \epsilon$, where $x_0$ and $x_1$ are drawn from uniform distribution $[-1, 1]$ and $\epsilon$ is drawn from uniform distribution $[-0.01, 0.01]$. The training set and test set sizes are 200 and 10,000, respectively. We use a very simple two-layer MLP structure with 2 inputs, 20 hidden neurons and 1 output.

We randomly initialized the model parameters and trained the model on the synthetic training dataset. As a control setup we also tried to freeze model weights in the first layer to prevent it from learning features. Note that a two layer MLP with the first layer frozen is equivalent to a linear regression model. That is, regression weights are learned on the second layer using fixed features extracted by the first layer. We plot the average GSNR of the second layer parameters for both the frozen and non-frozen cases. Figure 5 shows that in the non-frozen case, the average GSNR over parameters of

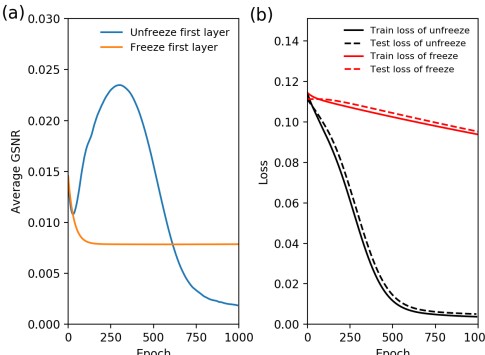 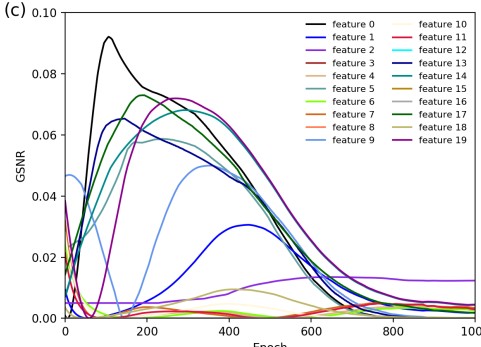

Figure 5: Average GSNR **(a)** and loss **(b)** curves for the frozen and non-frozen case. **(c)**: GSNR curves of individual parameters for the non-frozen case.

the second layer shows a significant upward process, whereas in the frozen case the average GSNR decreases in the beginning and remains at a low level during the whole training process.

In the non-frozen case, GSNR curve of individual parameters of the second layer are shown in Figure 5. The GSNR for some parameters show a significant upward process. To measure the quality of these features, we computed the Pearson correlation between them and the target output $y$, both at the beginning of training and at the maximum point of their GSNR curves. We can see that the learning process learns "good" features (high correlation value, *i.e.* with stronger correlation with $y$) from random initialized ones, as shown in Table 1. This shows that the GSNR increasing process is related to feature learning.

### 3.3 ANALYSIS OF TRAINING DYNAMICS BEHIND DNNS' GSNR BEHAVIOR

In this section, we will investigate the training dynamics behind the GSNR curve behavior. In the case of fully connected network structure, we can analytically show that the numerator of GSNR, *i.e.* the squared gradient mean of model parameters, tends to increase in the early training stage through feature learning.

Consider a fully connected network, whose parameters are $\theta = \{\mathbf{W}^{(1)}, \mathbf{b}^{(1)}, ..., \mathbf{W}^{(l_{max})}, \mathbf{b}^{(l_{max})}\}$, where $\mathbf{W}^{(1)}, \mathbf{b}^{(1)}$ are the weight matrix and bias of the first layer, and so on. We denote the activations of the $l$-th layer as $\mathbf{a}^{(l)} = \{a_s^{(l)}(\theta^{(l-)})\}$, where $s$ is the index for nodes/channels of this layer, and $\theta^{(l-)}$ is the collection of model parameters in the layers before $l$, *i.e.* $\theta^{(l-)} = \{\mathbf{W}^{(1)}, \mathbf{b}^{(1)}, ..., \mathbf{W}^{(l-1)}, \mathbf{b}^{(l-1)}\}$. In the forward pass on data sample $i$, $\{a_s^l(\theta^{(l-)})\}$ is multiplied by the weight matrix $\mathbf{W}^{(l)}$:

$$o_{i,c}^{(l)} = \sum_s W_{c,s}^{(l)} a_{i,s}^{(l)}(\theta^{(l-)}) \tag{26}$$

where $\mathbf{o}^{(l)} = \{o_{i,c}^{(l)}\}$ is the output of the matrix multiplication, for the $i$-th data sample, on the $l$-th layer, $c = \{1, 2, ..., C\}$ is the index of nodes/channels in the $(l+1)$-th layer. We use $\mathbf{g}_D^{(l)}$ to denote the average gradient of weights of the $l$-th layer $\mathbf{W}^{(l)}$, *i.e.* $\mathbf{g}_D^{(l)} = \frac{1}{n} \sum_{i=1}^n \frac{\partial L_i}{\partial \mathbf{W}^{(l)}}$, where $L_i$ is the loss of the $i$-th sample.

Here we show that the feature learning ability of DNNs plays a crucial role in the GSNR increasing process. More precisely, we show that the learning of features $\mathbf{a}^{(l)}(\theta^{(l-)})$, *i.e.* the learning of parameters $\theta^{(l-)}$ tends to increase the absolute value of $\mathbf{g}_D^{(l)}$. Consider the one-step change of gradient mean $\Delta \mathbf{g}_D^{(l)} = \mathbf{g}_{D,t+1}^{(l)} - \mathbf{g}_{D,t}^{(l)}$ with the learning rate $\lambda \to 0$. In one training step, $\theta$ is updated by $\Delta\theta = \theta_{t+1} - \theta_t = -\lambda \mathbf{g}_D(\theta)$. Using linear approximation with $\lambda \to 0$, we have

$$\Delta \mathbf{g}_{D,s,c}^{(l)} \approx \sum_j \frac{\partial \mathbf{g}_{D,s,c}^{(l)}}{\partial \theta_j} \Delta\theta_j = \sum_{\theta_j \in \theta^{(l-)}} \frac{\partial \mathbf{g}_{D,s,c}^{(l)}}{\partial \theta_j} \Delta\theta_j + \sum_{\theta_j \in \theta^{(l+)}} \frac{\partial \mathbf{g}_{D,s,c}^{(l)}}{\partial \theta_j} \Delta\theta_j \tag{27}$$

where $\theta^{(l-)}$ and $\theta^{(l+)}$ denote model parameters before and after the $l$-the layer (including the $l$-th), respectively.

We focus on the first term of eq. (27), *i.e.* the one-step change of $\mathbf{g}_D^{(l)}$ caused by learning $\theta^{(l-)}$. Substituting $\mathbf{g}_D^{(l)} = \frac{1}{n}\sum_{i=1}^{n} \frac{\partial L_i}{\partial \mathbf{W}^{(l)}}$ and $\Delta\theta_j = (-\lambda\frac{1}{n}\sum_{i=1}^{n}\frac{\partial L_i}{\partial\theta_j})$ into eq. (27), we have

$$\Delta\mathbf{g}_{D,s,c}^{(l)} = -\frac{\lambda}{n^2}\sum_{\theta_j \in \theta^{(l-)}} \mathbf{W}_{s,c}^{(l)}(\sum_{i=1}^{n} \frac{\partial L_i}{\partial o_{i,c}^{(l)}}\frac{\partial a_{i,s}^{(l)}}{\partial\theta_j})^2 + other\ terms \tag{28}$$

The detailed derivation of eq. (28) can be found in Appendix B. We can see the first term (which is a summation over parameters in $\theta^{(l-)}$) in eq. (28) has opposite sign with $\mathbf{W}_{s,c}^{(l)}$. This term will make $\Delta\mathbf{g}_{D,s,c}^{(l)}$ negatively correlated with $\mathbf{W}_{s,c}^{(l)}$. We plot the correlation between $\Delta\mathbf{g}_{D,s,c}^{(l)}$ with $\mathbf{W}_{s,c}^{(l)}$ for a model trained on MNIST for 200 epochs in Figure 6a. In the early training stage, they are indeed negatively correlated. For top-10% weights with larger absolute values, the negative correlation is even more significant.

Here we show that this negative correlation between $\Delta\mathbf{g}_{D,s,c}^{(l)}$ and $\mathbf{W}_{s,c}^{(l)}$ tends to increase the absolute value of $\mathbf{g}_D^{(l)}$ through an interesting mechanism. Consider the weights $\mathbf{W}_{s,c}^{(l)}$ with $\{\mathbf{W}_{s,c}^{(l)} > 0, \mathbf{g}_{D,s,c}^{(l)} < 0\}$. Learning $\theta^{l-}$ would decrease $\mathbf{g}_{D,s,c}^{(l)}$ and thus increase its absolute value because the first term in eq. (28) is negative. On the other hand, learning $\mathbf{W}_{s,c}^{(l)}$ would increase $\mathbf{W}_{s,c}^{(l)}$ and its absolute value because $\Delta\mathbf{W}_{s,c}^{(l)} = -\lambda\mathbf{g}_{D,s,c}^{(l)}$ is positive. This will form a positive feedback process, in which the numerator of GSNR, $(\mathbf{g}_{D,s,c}^{(l)})^2$, would increase and so is the GSNR. Similar analysis can be done for the case with $\{\mathbf{W}_{s,c}^{(l)} < 0, \mathbf{g}_{D,s,c}^{(l)} > 0\}$.

On the other hand, when $\{\mathbf{W}_{s,c}^{(l)}\mathbf{g}_{D,s,c}^{(l)} > 0\}$, we show that the weights tend to change into the earlier case, *i.e.* $\{\mathbf{W}_{s,c}^{(l)}\mathbf{g}_{D,s,c}^{(l)} < 0\}$ during training. Consider the case of $\{\mathbf{W}_{s,c}^{(l)} > 0, \mathbf{g}_{D,s,c}^{(l)} > 0\}$, the first term in eq. (28) is negative, learning $\theta^{(l-)}$ tends to decrease $\mathbf{g}_{D,s,c}^{(l)}$ or even change its sign. Another posibility is that learning $\mathbf{W}_{s,c}^{(l)}$ changes the sign of $\mathbf{W}_{s,c}^{(l)}$ because $\Delta\mathbf{W}_{s,c}^{(l)} = -\lambda\mathbf{g}_{D,s,c}^{(l)}$ is negative. In both cases the weights change into the earlier case with $\{\mathbf{W}_{s,c}^{(l)}\mathbf{g}_{D,s,c}^{(l)} < 0\}$. Similar analysis can be done for the case of $\{\mathbf{W}_{s,c}^{(l)} < 0, \mathbf{g}_{D,s,c}^{(l)} < 0\}$.

Therefore $\{\mathbf{W}_{s,c}^{(l)}\mathbf{g}_{D,s,c}^{(l)} < 0\}$ is a more stable state in the training process. For a simple model trained on MNIST, We plot the proportion of weights satisfying $\{\mathbf{W}_{s,c}^{(l)}\mathbf{g}_{D,s,c}^{(l)} < 0\}$ in Figure 6b and find that there are indeed more weights with $\{\mathbf{W}_{s,c}^{(l)}\mathbf{g}_{D,s,c}^{(l)} < 0\}$ than the opposite. Because weights with small absolute value easily change sign during training, we also plot this proportion for the top-10% weights with larger absolute values. We can see that for the weights with large absolute values, nearly 80% of them have opposite signs with their gradient mean, confirming our earlier analysis. For these weights, the numerator of GSNR, $(\mathbf{g}_{D,s,c}^{(l)})^2$, tends to increase through the positive feedback process as discussed above.

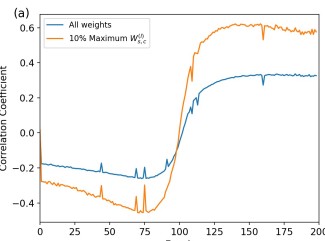 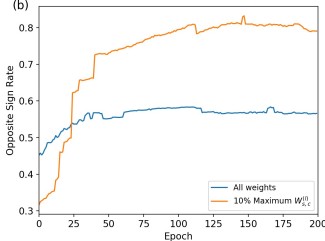

Figure 6: MNIST experiments. **Left**: Correlation between $\Delta\mathbf{g}_{D,s,c}^{(l)}$ and $\mathbf{W}_{s,c}^{(l)}$. **Right** : Ratio of weights that have opposite signs with their gradient mean.

Table 1: Pearson correlation between features and target output $y$, where $c_{t_0}$ and $c_{t_{max}}$ are correlations at the beginning of training and maximum of GSNR curve respectively.

| feature id | $c_{t_0}$ | $c_{t_{max}}$ |
|---|---|---|
| 0 | -0.11 | 0.47 |
| 5 | 0.11 | 0.44 |
| 13 | 0.07 | 0.40 |
| 14 | -0.21 | -0.27 |
| 17 | -0.33 | 0.53 |

## 4 SUMMARY

In this paper, we performed a series of analysis on the role of model parameters' GSNR in deep neural networks' generalization ability. We showed that large GSNR is a key to small generalization gap, and gradient descent training naturally incurs and exploits large GSNR as the model discovers useful features in learning.

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

# A APPENDIX A

## A.1 MODEL STRUCTURE IN SECTION 2.4

As shown in Table 2, all models in the experiment consist of 2 Conv-Relu-MaxPooling blocks and 2 fully-connected layers, but they are different in the number of channels. We choose the number of channels $p$ from $\{6, 8, 10, 12, 14, 16, 18, 20\}$.

Table 2: Model structure On MNIST in Section 2.4. $p$ is the number of channels and $q = int(2.5 * p)$

| Layer | input #channels | output #channels |
|---|---|---|
| conv + relu + maxpooling | 1 | $p$ |
| conv + relu + maxpooling | $p$ | $q$ |
| flatten | - | - |
| fc + relu | 16 * $q$ | 10 * $q$ |
| fc + relu | 10 * $q$ | 10 |
| softmax | - | - |

## A.2 EXPERIMENT ON CIFAR10

Different from the experiment on MNIST, we use a deeper network on CIFAR10. We also include the Batch Normalization (BN) layer, because we find that it's difficult for the network to converge in the absence of it. The network consists of 4 Conv-BN-Relu-Conv-BN-Relu-MaxPooling blocks and 3 fully-connected layers. More details are shown in Table 3.

Table 3: Model structure on CIFAR10. $p$ is the number of channels.

| Layer | input #channels | output #channels |
|---|---|---|
| conv + bn + relu | 3 | $p$ |
| conv + bn + relu | $p$ | $p$ |
| maxpooling | - | - |
| conv + bn + relu | $p$ | $2p$ |
| conv + bn + relu | $2p$ | $2p$ |
| maxpooling | - | - |
| conv + bn + relu | $2p$ | $4p$ |
| conv + bn + relu | $4p$ | $4p$ |
| maxpooling | - | - |
| conv + bn + relu | $4p$ | $8p$ |
| conv + bn + relu | $8p$ | $8p$ |
| maxpooling | - | - |
| flatten | - | - |
| fc + relu | 32 * $q$ | 8 * $q$ |
| fc + relu | 8 * $q$ | 8 * $q$ |
| fc | 8 * $q$ | 10 |
| softmax | - | - |

The experiment is conducted under a similar setting as that of MNIST in section 2.4. We choose $n \in \{2000, 4000, 6000, 8000, 10000\}$, $p_{random} \in \{0.0, 0.2, 0.4\}$, $ch \in \{6, 8, 10, 12, 14, 16, 18\}$. We use the gradient descent training (Not SGD), with a small learning rate of $0.001$. The left and right hand sides of 19 at different epochs are shown in Figure 7, where each point represents one specific combination of the above settings. Note that at the evaluation step of every epoch, we use

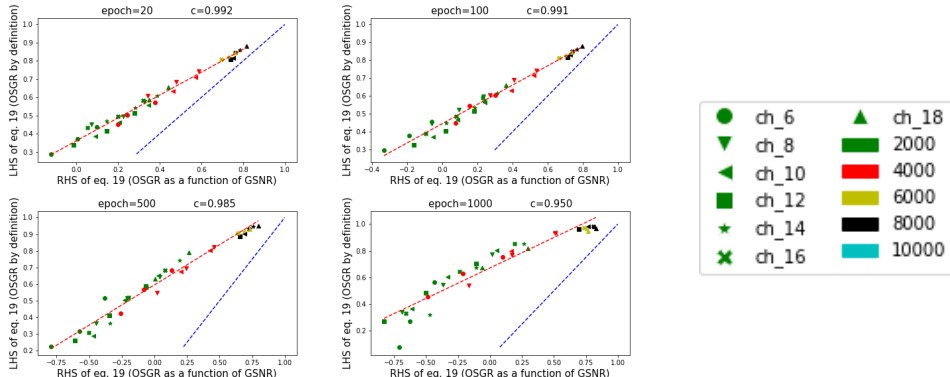

Figure 7: Left hand (LHS) and right side (RHS) of eq. (19). Points are drawn under different experiment settings. Left figure: LHS vs RHS relation at epoch 20, 100, 500, 1000.

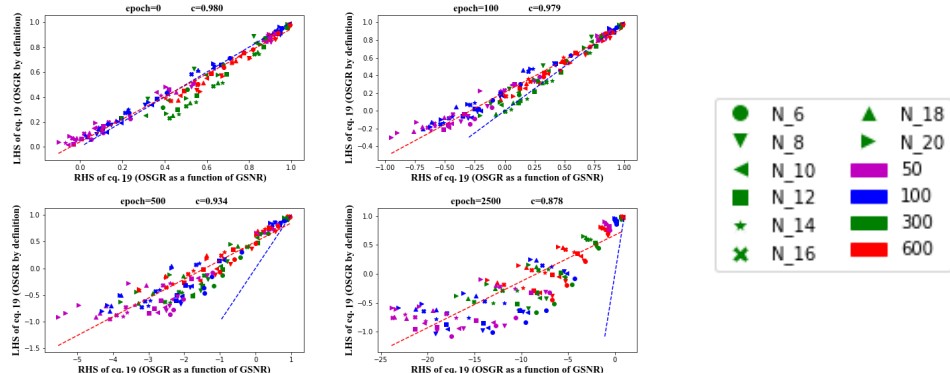

Figure 8: Similar with Fig. 3, but for a toy regression model discussed in in Appendix A.3.

the same mean and variance inside the BN layers as the training dataset. That's to ensure that the network and loss function are consistent between training and test.

At the beginning of training, compared to that of MNIST, the data points no longer perfectly resides on the diagonal dashed line. We suppose that's beacuse of the presence of BN layer, whose internal parameters, *i.e.* running mean and running variance, are not regular learnable parameters in the optimization process, but change their values in a different way. Their change affects the OSGR, yet we could not include them in the estimation of OSGR. However, the strong positive correlation between the left and right hand sides of eq. (19) can always be observed until the training begins to converge.

## A.3   EXPERIMENT ON TOY DATASET

In this section we show a simple two-layer regression model consists of a FC-Relu structure with only 2 inputs, 1 hidden layer with $N$ neurons and 1 output. A similar synthetic dataset with the training data used in the experiment of Section 3.2 is generated as follows. Each data point is constructed *i.i.d.* using $y = x_0 x_1 + \epsilon$, where $x_0$ and $x_1$ are drawn from uniform distribution of $[-1, 1]$ and $\epsilon$ is drawn from uniform distribution of $[-\eta_{noise}, \eta_{noise}]$.

To estimate eq. (24), we randomly generate 100 training sets with $n$ samples each, *i.e.* $M$=100, and a test set with 20,000 samples. To cover different conditions, we (1) choose $n \in \{50, 100, 300, 600, 1000, 2000, 6000\}$; (2) inject noise with $\eta_{noise} \in \{0.2, 2, 4, 6, 8\}$; (3) perturb model structures by choosing $N \in \{6, 8, 10, 12, 14, 16, 18, 20\}$. We use gradient descent with learning rate of 0.001.

Figure 8 shows a similar behavior as Fig. 3. During the early training stages, the LHS and RHS of eq. (19) are very close. Their highly correlated relation remains until training converges, whereas the RHS of eq. (19) decreases significantly.

## B  APPENDIX B

Derivation of eq. (28)

$$\Delta \mathbf{g}_{D,s,c}^{(l)} = \sum_{\theta_j \in \theta^{(l-)}} \frac{\partial \mathbf{g}_{D,s,c}^{(l)}}{\partial \theta_j} \Delta \theta_j + other\ terms \tag{29}$$

$$= \sum_{\theta_j \in \theta^{(l-)}} \frac{\partial (\frac{1}{n} \sum_{i=1}^{n} \frac{\partial L_i}{\partial \mathbf{W}_{s,c}^{(l)}})}{\partial \theta_j} (-\lambda \frac{1}{n} \sum_{i=1}^{n} \frac{\partial L_i}{\partial \theta_j}) + other\ terms \tag{30}$$

$$= \sum_{\theta_j \in \theta^{(l-)}} \frac{\partial (\frac{1}{n} \sum_{i=1}^{n} \frac{\partial L_i}{\partial o_{i,c}^{(l)}} \frac{\partial o_{i,c}^{(l)}}{\partial \mathbf{W}_{s,c}^{(l)}})}{\partial \theta_j} (-\frac{\lambda}{n} \sum_{i=1}^{n} \sum_{s',c'} \frac{\partial L_i}{\partial o_{i,c'}^{(l)}} \frac{\partial o_{i,c'}^{(l)}}{\partial a_{i,s'}^{(l)}} \frac{\partial a_{i,s'}^{(l)}}{\partial \theta_j}) + other\ terms \tag{31}$$

$$= -\frac{\lambda}{n^2} \sum_{\theta_j \in \theta^{(l-)}} \frac{\partial (\sum_{i=1}^{n} \frac{\partial L_i}{\partial o_{i,c}^{(l)}} a_{i,s}^{(l)})}{\partial \theta_j} (\sum_{i=1}^{n} \sum_{s',c'} \frac{\partial L_i}{\partial o_{i,c'}^{(l)}} \mathbf{W}_{s',c'}^{(l)} \frac{\partial a_{i,s'}^{(l)}}{\partial \theta_j}) + other\ terms \tag{32}$$

$$= -\frac{\lambda}{n^2} \sum_{\theta_j \in \theta^{(l-)}} \sum_{i=1}^{n} (\frac{\partial L_i}{\partial o_{i,c}^{(l)}} \frac{\partial a_{i,s}^{(l)}}{\partial \theta_j} + \frac{\partial^2 L_i}{\partial o_{i,c}^{(l)} \partial \theta_j} a_{i,s}^{(l)})(\sum_{s',c'} \mathbf{W}_{s',c'}^{(l)} \sum_{i=1}^{n} \frac{\partial L_i}{\partial o_{i,c'}^{(l)}} \frac{\partial a_{i,s'}^{(l)}}{\partial \theta_j})$$

$$+ other\ terms \tag{33}$$

Above we used $\frac{\partial o_{i,c'}^{(l)}}{\partial a_{i,s'}^{(l)}} = \mathbf{W}_{s',c'}^{(l)}$ and $\frac{\partial o_{i,c}^{(l)}}{\partial \mathbf{W}_{s,c}^{(l)}} = a_{i,s}^{(l)}$ that can both be derived from eq. (26). Consider the first term of eq. (33). When $s' = s, c' = c$, we have

$$\Delta \mathbf{g}_{s,c}^{(l)} = -\frac{\lambda}{n^2} \sum_{\theta_j \in \theta^{(l-)}} \mathbf{W}_{s,c}^{(l)} (\sum_{i=1}^{n} \frac{\partial L_i}{\partial o_{i,c}^{(l)}} \frac{\partial a_{i,s}^{(l)}}{\partial \theta_j})^2 + other\ terms \tag{34}$$

Note that the term related to $\frac{\partial^2 L_i}{\partial o_{i,c}^{(l)} \partial \theta_j} a_{i,s}^{(l)}$ and the terms when $s' \neq s$ or $c' \neq c$ in eq. (33) are merged into *other terms* of eq. (34).

## C   APPENDIX C

**Notations**

| | |
|---|---|
| $\mathcal{Z}$ | A data distribution satisfies $\mathcal{X} \times \mathcal{Y}$ |
| $s$ or $(x, y)$ | A single data sample |
| $D$ | Training set consists of $n$ samples drawn from $\mathcal{Z}$ |
| $D'$ | Test set consists of $n'$ samples drawn from $\mathcal{Z}$ |
| $\theta$ | Model parameters, whose components are denoted as $\theta_j$ |
| $\mathbf{g}_s(\theta)$ or $\mathbf{g}_i(\theta)$ | Parameters' gradient *w.r.t.* a single data sample $s$ or $(x_i, y_i)$ |
| $\tilde{\mathbf{g}}(\theta)$ | Mean values of parameters' gradient over a total data distribution, *i.e.*, $\mathrm{E}_{s\sim\mathcal{Z}}(\mathbf{g}_s(\theta))$ |
| $\mathbf{g}_D(\theta)$ | Average gradient over the training dataset, *i.e.*, $\frac{1}{n}\sum_{i=1}^n \mathbf{g}_i(\theta)$ |
| $\mathbf{g}_{D'}(\theta)$ | Average gradient over the test dataset, *i.e.*, $\frac{1}{n'}\sum_{i=1}^{n'} \mathbf{g}'_i(\theta)$. Note that, in eq. (5), we assume $n' = n$ |
| $\mathbf{g}_{D,j}$ | Same as $\mathbf{g}_D(\theta_j)$ |
| $\rho^2(\theta)$ | Variance of parameters' gradient of a single sample, *i.e.*, $\mathrm{Var}_{s\sim\mathcal{Z}}(\mathbf{g}_s(\theta))$ |
| $\rho_j^2$ | Same as $\rho^2(\theta_j)$ |
| $\sigma^2(\theta)$ | Variance of the average gradient over a training dataset of size $n$, *i.e.*, $\mathrm{Var}_{D\sim\mathcal{Z}^n}[\mathbf{g}_D(\theta)]$ |
| $\sigma_j^2$ | Same as $\sigma^2(\theta_j)$ |
| $r_j$ or $r(\theta_j)$ | Gradient signal to noise ratio (GSNR) of model parameter $\theta_j$ |
| $L[D]$ | Empirical training loss, *i.e.*, $\frac{1}{n}\sum_{i=1}^n L(y_i, f(x_i, \theta))$ |
| $L[D']$ | Empirical test loss, *i.e.*, $\frac{1}{n'}\sum_{i=1}^{n'} L(y'_i, f(x'_i, \theta)))$ |
| $\Delta L[D]$ | One-step training loss decrease |
| $\Delta L_j[D]$ | One-step training loss decrease caused by updating one parameter $\theta_j$ |
| $\mathbf{R}(\mathcal{Z}, n)$ | One-step generalization ratio (OSGR) for the training and test sets of size $n$ sampled from data distribution $\mathcal{Z}$, *i.e.*, $\frac{E_{D,D'\sim\mathcal{Z}^n}(\Delta L[D'])}{E_{D\sim\mathcal{Z}^n}(\Delta L[D])}$ |
| $\lambda$ | Learning rate |
| $\bigtriangledown$ | One-step generalization gap increment, *i.e.*, $\Delta L[D]$ - $\Delta L[D']$ |
| $\epsilon$ | Random variables with zero mean and variance $\sigma^2(\theta)$ |
| $\mathbf{W}^{(l)}$ and $\mathbf{b}^{(l)}$ | Model parameters (weight matrix and bias) of the $l$-th layer |
| $\theta^{(l-)}$ | Collection of model parameters over all the layers before the $l$-th layer |
| $\mathbf{g}_D^{(l)}$ | Average gradient of $\mathbf{W}^{(l)}$ over the training dataset |
| $\theta^{(l+)}$ | Collection of model parameters over all the layers after the $l$-th layer, including the $l$-th layer |
| $\mathbf{a}^{(l)} = \{a_s^{(l)}(\theta^{(l-)})\}$ | Activations of the $l$-th layer, where $s = \{1, 2, ..., S\}$ is the index of nodes/channels in the $l$-th layer. |
| $\mathbf{o}^{(l)} = \{o_c^{(l)}\}$ | Outputs of matrix multiplication of the $l$-th layer, where $c = \{1, 2, ..., C\}$ is index of nodes/channels in the $(l + 1)$-th layer. |
| $a_{i,s}^{(l)}$ and $o_{i,c}^{(l)}$ | $a_s^{(l)}$ and $o_c^{(l)}$ evaluated on data sample $i$ |

