# OpenReview forum: "Understanding Why Neural Networks Generalize Well Through GSNR of Parameters"
_ICLR.cc/2020/Conference — Accept (Spotlight)_

### Official Review · AnonReviewer1 · 2019-10-22
**Official Blind Review #1**

**Rating:** 6

**Review:**

In this work, the authors suggest a new point of view on generalization through the lens of the distribution of the per-sample gradients. The authors consider the variance and mean of the per-sample gradients for each parameter of the model and define for each parameter the Gradient Signal to Noise ratio (GSNR). The GSNR of a parameter is the ratio between the mean squared of the gradient per parameter per sample (computed over the samples) and the variance of the gradient per parameter per sample (also computed over the samples). The GSNR is promising as a measure of generalization and the authors provide a nice leading order derivation of the GSNR as a proxy for the measure of the generalization gap in the model. After the derivation, experimental results on MNIST are presented and suggest that empirically there is a relation between the generalization gap of neural network trained by gradient descent and the GSNR quantity given in the paper. Next the author analyze the GNSR of DNNs as opposed with shallow models or other learning techniques and observe that the GSNR differs when using random labels (lower GSNR) as compared with true labels and exhibits different behavior along training for DNNs and gradient descent.

 Assumption 2.3.1 is not necessarily the most realistic in the current training paradigm since multi-epoch training indeed separates the training and testing per-sample gradient distributions significantly.

Overall the paper gives a fresh (as far as I know) and nice idea on generalization of neural networks. The derivation requires quite a few stringent assumption (the leading order analysis, on the step size, assumption 2.3.1 on the test and train gradient distributions) the experiments do suggest that the theory is valid to an extent, especially during the early parts of training. In contrast, the presentation of the paper distracts from the work and needs additional cleaning up. Also, the experimental section 2.4 would benefit additional empirical analysis on other datasets  and additional experiments, as well as more thorough explanation of the experiments. The writing of the paper needs additional proofreading as currently it is easy to spot typos and grammar errors along the paper. I currently vote weak reject, for solid content and not-quite-ready presentation.

Additional experiments and careful proofreading should definitely enhance the paper and get it to the level of publications, so I am willing to change my decision if the authors improve the writing and the overall presentation. I like the idea presented in the paper and encourage the authors to resubmit a more tidy draft.

smaller presentation issues and typos/grammar issues:

Figure 1: name X and Y labels with more meaningful names instead of RHS,LHS
Figure 3(c): add legend
line 64: consists >>> which consists
line 73: we refer R(z) >>> we refer to R(z)
line 79 futher >>> further overall sentence needs more work
line 92 distribution becomes >>> distributions become
line 100: using common notation summing over n samples, the sum should possibly start from i=1 to n instead of i = 0 to n
line 132 include >>> includes (also possibly rephrase sentence)
line 136 vefity >>> verify
line 137: M number >>> M
line 157 the data points closely distributed >>> the data points are closely distributed
line 162 thorough analytically >>> thorough analytical




-------- Update After Rebuttal --------

I have read the comments the authors made and reviewed the paper's revision.
The presentation has improved a lot (Even though there is still room to go).
Nonetheless at this time I choose to update my score to a weak accept, as I think the authors bring a fresh (as far as I know) and interesting idea with regards to empirical generalization.

**Experience Assessment:**

I have read many papers in this area.

**Review Assessment: Checking Correctness Of Derivations And Theory:**

I carefully checked the derivations and theory.

**Review Assessment: Checking Correctness Of Experiments:**

I assessed the sensibility of the experiments.

**Review Assessment: Thoroughness In Paper Reading:**

I read the paper thoroughly.

---

> ### Author Response · Authors · 2019-11-13
> **Response to Review #1**
>
> Thank you for your responsible review and comments.
>
> We have resubmitted a revision and keep improving the overall presentation, where
> (1) the paper has been largely revamped to improve its quality of presentation;
> (2) notations are changed to improve readability and consistency, and a notation table is added in Appendix C;
> (3) all figures are re-generated to make them more readable.
> (4) additional experiments on CIFAR10 and toy dataset are included.
>
> We conducted the experiments on CIFAR10 and toy dataset, in Appendix A.2 and A.3 respectively. Briefly, for CIFAR10 we used a deeper CNN with batch normalization layer, and for the toy dataset which is constructed by a very simple way, we used a 2-layers MLP. The experiment settings on both dataset are similar to that on MNIST, and we got similar observations.
>
> Assumption 2.3.1 usually holds true in the early training stage. Multi-epoch training undoubtedly separates the training and testing per-sample gradient distributions significantly, which will break the stringent Assumption 2.3.1 in the late training stage. However, the experiments on Toy-DNN, MNIST and CIFAR10 together indicate that, for common network structures (MLP or CNN), with the commonly used step size of 0.001, the strongly positive correlation of LHS and RHS of equation (19)(in the revision) remains until training converges even when the Assumption 2.3.1 no longer holds. Therefore, the relation that “the larger GSNR during training leads to better generalization ability” is valid during the whole training process with or without Assumption 2.3.1. We have also improved presentation of Assumption 2.3.1 to make it more intuitive.

---

### Official Review · AnonReviewer3 · 2019-10-27
**Official Blind Review #3**

**Rating:** 3

**Review:**

The paper defines the quantity of "gradient SNR" (GSNR), shows that larger GSNR leads to better generalization, and shows that SGD training of deep networks has large GSNR. It tells a great story on why SGD-trained DNNs have good generalization.

This topic is highly relevant to this conference.

However, I struggle to rate this paper, since I feel swamped with math. It is hard work to read this paper, and I can honestly say that I could semi-confidently follow until about Eq. (8). To even get there, I had to scroll back and forth to remember the definitions of the various symbols. The math may be very well correct, but it is infeasible to verify (or follow) it fully. It does not make it easier that one cannot really search a PDF for greek symbols with indices etc. Someone who reads theoretical papers all day long might do better here.

This is the reason I rate the paper Weak Reject.

Some feedback points:

Section 2.1:

Eq. (1): It seems the common definition of SNR is the ratio of mean standard deviation. Your SNR is its square. This should be explained.

I think it would help the reader a lot to give some intuitive meaning to the GSNR value. Can you, in Section 2.1, explain with examples what typical (or extreme) values would be?

Assumption 2.3.1:

This is dropped on the reader without any motivation. It is also confusing: "we will make our derivation under the non-overfitting limit approximation" conflicts with "In the early training stage,..." So is this whole derivation only true in the early stages?

Assumption 2.3.1 seems to address a thought I had when reading this: At the end of the training, I would expect mu_q(theta) to be zero (the definition of convergence). At the start, it is arbitrary as it entirely depends on the initial values. So this paper must look at some part between the two extremes to make sense. Is it? Is this assumption related?

What is the difference between \sigma and \rho? Seems one is on the data distribution and one on a sampled set. But then why is \mu the same in both cases (Eq. (1) vs. Eq. (5))?

All plots:

The plot labels are far too small to be readable.

**Experience Assessment:**

I have read many papers in this area.

**Review Assessment: Checking Correctness Of Derivations And Theory:**

I assessed the sensibility of the derivations and theory.

**Review Assessment: Checking Correctness Of Experiments:**

I assessed the sensibility of the experiments.

**Review Assessment: Thoroughness In Paper Reading:**

I read the paper at least twice and used my best judgement in assessing the paper.

---

> ### Author Response · Authors · 2019-11-13
> **Response to Review #3**
>
> Thank you for your responsible review and comments.
>
> We have resubmitted a revision and keep improving the overall presentation, where
> (1) the paper has been largely revamped to improve its quality of presentation;
> (2) notations are changed to improve readability and consistency and a notation table is added in Appendix C;
> (3) all figures are re-generated to make them more readable.
> (4) additional experiments on CIFAR10 and toy dataset are included.
>
> Your main concerns are addressed below.
>
> The definition of SNR on Wikipedia is the ratio between squared mean and variance, which ensures it is positive.
>
> Assumption 2.3.1 is needed for the derivation. It usually holds true in the early training stage. Multi-epoch training undoubtedly separates the training and testing per-sample gradient distributions significantly, which will break the stringent Assumption 2.3.1 in the late training stage. However, the experiments on Toy-DNN, Mnist and CIFAR10 together indicate that with the commonly used step size of 0.001, the strongly positive correlation of LHS and RHS of equation(19) (in the revision) remains until training converges even when the Assumption 2.3.1 no longer holds. Therefore, the relation that “the larger GSNR during training leads to better generalization ability” is valid during the whole training process with or without Assumption 2.3.1. We have also improved presentation of Assumption 2.3.1 to make it more intuitive.
>
> We have \sigma represents “the standard deviation of gradient mean over a training dataset of size n” whereas \rho represents “the standard deviation of parameters’ gradient of a single sample”. Under Assumption 2.3.1, it can be proved that they differ by a factor of n, which is the dataset size. \mu is the same, please refer to equation 7 in the first version or equation 6 in the revision. We give a proof here.
> $$E_{D\sim \mathcal Z^n}[\mathbf g_D(\mathbf\theta)]=E_{(x_0,y_0)\sim \mathcal Z,(x_1,y_1)\sim \mathcal Z,...,(x_n,y_n)\sim \mathcal Z}(\frac1n\sum_{i=1}^n\mathbf g_i(\mathbf \theta))=\frac1n\sum_{i=1}^nE_{(x_i,y_i)\sim \mathcal Z}(\mathbf g_i(\mathbf \theta))=E_{(x,y)\sim \mathcal Z}(\mathbf g(x,y,\mathbf \theta))\equiv\tilde{\mathbf g}(\mathbf\theta)$$

---

### Official Review · AnonReviewer4 · 2019-11-22
**Official Blind Review #4**

**Rating:** 6

**Review:**

Paper Summary

This paper introduces a quantity termed the "one-step generalization ratio" . They derive approximate relations between OSGR and GSNR then show experimental results demonstrating the validity of these approximations, thus linking GSNR and a quantity related to generalization. They investigate the empirical value of GSNR during training of a neural network on Cifar10 with real labels vs. random labels. The final section derives a relation which attempts to explain the correlation between the size of the expected gradient and the learning of features.

Review

First, I should note that I am not well-versed in this specific line of work (GSNR ,as other papers were cited with this concept), and thus I cannot readily assess whether the contributions are "well-placed in the literature".

For the derivations and theory, I checked as much of the derivations as possible, although I'm not familiar with all the assumptions used throughout, but they seem relatively reasonable and supported by the experiments. I.e. 2.3.1 seems reasonable at the start of training. But I'm not sure how to evaluate some of the limiting arguments, like taking the step size to zero.

The paper addresses an important topic and is interesting. The arguments seem well reasoned and logical. The experiments addressed the veracity of of the approximate relations derived in the first section as well as some interesting trends related to the GSNR quantity during training of some simple models.

Small Concerns

On line 188 it is mentioned that GSNR is the "reason" for a particular phenomenon or explains a phenomenon. It certainly seems reasonable that it is correlated or an indicator of the underlying phenomenon, but it doesn't seem (at least intuitively) that GSNR is the cause or reason for the generalization concepts being examined.


**Experience Assessment:**

I do not know much about this area.

**Review Assessment: Checking Correctness Of Derivations And Theory:**

I assessed the sensibility of the derivations and theory.

**Review Assessment: Checking Correctness Of Experiments:**

I assessed the sensibility of the experiments.

**Review Assessment: Thoroughness In Paper Reading:**

I read the paper at least twice and used my best judgement in assessing the paper.

---

### Author Response · Authors · 2019-11-14
**We have submitted a second revision.**

Hi, we have submitted a second revision where
(1) we changed "eq. 17" to "eq. 19" in the plot labels of  Figure 1, Figure 5 and Figure 6;
(2) we removed the definitions of expectation of g_D and g_D' from  the beginning of section 2.3, to improve the readability.

---

### Author Response · Authors · 2019-11-15
**We have submitted a final revision of the Discussion Stage 2.**

Hi, we have submitted the final revision. Changes in this revision are mainly about spelling errors.
Thanks.

---

### Decision · Program_Chairs · 2019-12-19

**Decision:**

Accept (Spotlight)

**Comment:**

Quoting a reviewer for a very nice summary:

"In this work, the authors suggest a new point of view on generalization through the lens of the distribution of the per-sample gradients. The authors consider the variance and mean of the per-sample gradients for each parameter of the model and define for each parameter the Gradient Signal to Noise ratio (GSNR). The GSNR of a parameter is the ratio between the mean squared of the gradient per parameter per sample (computed over the samples) and the variance of the gradient per parameter per sample (also computed over the samples). The GSNR is promising as a measure of generalization and the authors provide a nice leading order derivation of the GSNR as a proxy for the measure of the generalization gap in the model."

The majority of the reviewers vote to accept this paper. We can view the 3 as a weak signal as that reviewer stated in his review that he struggled to rate the paper because it contained a lot of math.